# RL-PMO: A Reinforcement Learning-Based Optimization Algorithm for Parallel SFC Migration

**DOI:** 10.3390/s26010242

**Published:** 2025-12-30

**Authors:** Hefei Hu, Zining Liu, Fan Wu

**Affiliations:** School of Information and Communication Engineering, Beijing University of Posts and Telecommunications, Beijing 100876, China; huhefei@bupt.edu.cn (H.H.); liuzining@bupt.edu.cn (Z.L.)

**Keywords:** network function virtualization, service function chain, offline reinforcement learning, parallel migration

## Abstract

In edge networks, hardware failures and resource pressure may disrupt Service Function Chains (SFCs) deployed on the failed node, making it necessary to efficiently migrate multiple Virtual Network Functions (VNFs) under limited resources. To address these challenges, this paper proposes an offline reinforcement learning-based parallel migration optimization algorithm (RL-PMO) to enable parallel migration of multiple VNFs. The method follows a two-stage framework: in the first stage, improved heuristic algorithms are used to generate high-quality migration trajectories and construct a multi-scenario dataset; in the second stage, the Decision Mamba model is employed to train the policy network. With its selective modeling capability for structured sequences, Decision Mamba can capture the dependencies between VNFs and underlying resources. Combined with a twin-critic architecture and CQL regularization, the model effectively mitigates distribution shift and Q-value overestimation. The simulation results show that RL-PMO maintains approximately a 95% migration success rate across different load conditions and improves performance by about 13% under low and medium loads and up to 17% under high loads compared with typical offline RL algorithms such as IQL. Overall, RL-PMO provides an efficient, reliable, and resource-aware solution for SFC migration in node failure scenarios.

## 1. Introduction

With the deep integration of 5G technology and edge computing, Network Function Virtualization (NFV) has leveraged its core advantage of “software-defined network functions” to decouple network functionalities from dedicated hardware. This paradigm shift has significantly transformed network architecture and has become a pivotal technology for enabling flexible deployment and dynamic scheduling in edge networks. Within the NFV framework, a Service Function Chain (SFC) connects multiple Virtual Network Functions (VNFs) in a predefined order according to service requirements, thereby forming a flexible delivery path for customized network services [1]. In representative edge-computing scenarios such as the Internet of Things (IoT) and sensor networks, large volumes of real-time sensing data must be processed by VNFs deployed at edge nodes; as a result, SFCs serve as a crucial component of the processing workflow.

However, edge nodes are often constrained by complex deployment environments and limited hardware resources, making them highly susceptible to SFC disruptions caused by hardware failures, resource overload, or link congestion. Once an edge node hosting VNFs fails, all associated SFCs become paralyzed, leading to degraded Quality of Service (QoS) or even complete service outages. In Internet of Things (IoT) and sensor-network environments characterized by real-time sensing-data processing, such interruptions further compromise the continuity and accuracy of critical sensing tasks. Therefore, maintaining the continuity of SFCs is essential for ensuring the real-time performance and reliability of IoT and sensor-network systems.

To maintain SFC continuity under node failures, dynamic VNF migration has emerged as a key solution. Existing dynamic migration methods fall into two categories: serial migration and parallel migration. Serial migration transfers VNFs one at a time and effectively avoids resource conflicts but incurs considerable delay, making it unsuitable for latency-sensitive services. In contrast, parallel migration transfers multiple VNFs simultaneously to reduce overall migration latency. However, parallel migration is prone to higher failure rates due to resource contention during the migration process.

Based on the above background, this paper focuses on the failure-driven parallel migration of multiple VNFs. When a physical edge node fails, all Virtual Network Functions (VNFs) deployed on that node must be migrated simultaneously to other healthy nodes within a limited time to ensure the service continuity of the affected Service Function Chains (SFCs). This migration process must satisfy three core objectives: (i) maximizing the migration success rate; (ii) minimizing the migration time; and (iii) controlling the migration cost.

Under resource competition, bandwidth coupling, and multiple constraints, jointly optimizing these objectives becomes a typical NP-hard combinatorial problem. Therefore, this paper adopts an offline reinforcement learning (offline RL) approach. By generating diverse and safe migration trajectories using heuristic algorithms in a simulation environment, the model can learn globally coordinated migration strategies. Furthermore, to improve decision efficiency and representation capability, we employ the efficient sequence model Decision Mamba as the policy network to better capture the dependencies between VNFs and underlying resources.

The main contributions of this paper can be summarized as follows:Modeling the migration of all VNFs on the failed node as a single-step multi-objective optimization problem enables the system to maintain a high migration success rate while keeping the overall migration delay and resource overhead within acceptable limits.This paper designs an improved CSSA–PSO hybrid heuristic method to generate diverse and feasible migration trajectories under various failure patterns and load levels.To capture the dependencies among VNFs, network resources, and migration constraints, we adopt Decision Mamba as the policy network. In addition, we incorporate a twin-critic architecture together with a Conservative Q-Learning (CQL) regularization term to mitigate the inherent overestimation and distribution shift issues in offline reinforcement learning.

## 2. Related Work

In the integrated framework of Network Function Virtualization (NFV) and edge computing, dynamic migration of Service Function Chains (SFCs) plays a crucial role in maintaining the continuity of network services. Existing studies can be broadly categorized into performance-oriented migration and failure-recovery-oriented migration. While the former remains the mainstream direction, the latter continues to face challenges due to the unpredictable nature of edge node failures and stringent resource constraints. As a result, recent research has increasingly shifted toward failure-prediction-driven proactive migration.

For instance, Wang et al. [2] proposed a log-driven node failure prediction method that extracts critical system features from operation logs and employs machine learning models to enhance prediction accuracy, thereby supporting fault recovery and migration decision-making. Zhai et al. [3] proposed an enhanced LSTM model optimized by SA-PSO to identify high-risk servers and used an improved Sparrow Search Algorithm (ISSA) to perform proactive SFC migration, reducing migration cost and latency. Similarly, Kabdjou and Shinomiya [4] employed an LSTM model optimized by Super-SAPSO for failure prediction in MEC environments and designed a robust migration scheme that accounts for node security levels and user mobility. Building on these studies, this work focuses on VNF migration after a potential failure node has been predicted.

Traditional SFC migration and resource allocation approaches commonly rely on heuristic algorithms such as greedy selection or rule-based strategies. For example, the work in [5] proposed a prediction-based VNF migration mechanism that determines migration timing and target nodes through heuristic rules to mitigate performance degradation; the study in [6] introduced the VNF-RM algorithm, which employs a delay-aware heuristic strategy to sequentially select the migration order and destination nodes, thereby minimizing end-to-end latency; and [7] developed a migration cost model and adopted a greedy heuristic method to reduce additional overhead while ensuring service continuity.

Most of the above methods rely on serial migration. When multiple VNFs need to be migrated simultaneously, such a serial strategy leads to a nearly linear increase in migration time. Recently, studies such as [3,4] have begun exploring parallel migration, which schedules multiple VNF tasks concurrently to better utilize available link bandwidth and reduce queuing delay, thereby substantially improving overall migration efficiency.

Despite their practical feasibility and ease of engineering implementation, parallel heuristic algorithms still exhibit two inherent limitations: (1) Lack of global optimization capability. This limitation becomes more pronounced in multi-VNF parallel migration scenarios, where accurately modeling resource contention and coupling among VNFs is highly challenging. The problem is even more evident in sensor-network and IoT-edge environments driven by real-time sensing-data processing, where traffic patterns and resource usage are highly dynamic, making heuristic methods more prone to local optima. (2) Lack of learning ability and poor adaptability to dynamic environments, which makes it difficult for such methods to maintain stable migration performance when sensing data fluctuates frequently.

To overcome these limitations, researchers have explored data-driven and multi-VNF parallel migration strategies. In recent years, Deep Reinforcement Learning (DRL) has been increasingly applied to SFC migration to enable adaptive, intelligent decision-making. For instance, ref. [8] formulated migration as a Markov Decision Process (MDP) and used online RL for dynamic adjustment of VNF scaling and relocation. Ref. [9] adopted a deep Q-learning strategy to iteratively refine migration decisions and minimize service interruption, while ref. [10] proposed an interference-aware migration algorithm balancing throughput and cost. In addition, ref. [11] introduced a multi-agent DRL framework that coordinates multiple agents for joint migration of multiple SFCs, and ref. [12] proposed a cluster-based migration scheme that treats tightly coupled VNFs as a single migration unit to reduce embedding cost and satisfy latency constraints. These studies partially address the limitations of serial migration, but their parallelism remains limited to subsets or clusters of VNFs.

However, online reinforcement learning is not suitable for failure-driven multi-VNF parallel migration. Its trial-and-error exploration generates many unsafe decisions during early training. At the same time, failure scenarios require migration decisions within very short time windows, while online RL cannot converge within such limited time. This limitation is particularly critical in sensor-network and other latency-sensitive edge environments, where any unstable decision may directly interrupt sensing tasks, further amplifying the risks and infeasibility of online RL.

In recent years, Offline RL has demonstrated strong performance in robotics [13] and autonomous driving [14], where it can learn robust and interpretable decision policies from pre-collected datasets while ensuring operational safety. In networking systems, Offline RL has also begun to attract attention. For example, the work in [15] used Offline RL to build a bandwidth estimation model that reduces reliance on real-time probing and interaction, demonstrating the feasibility of Offline RL for network QoS assurance and bandwidth scheduling.

Compared with existing studies, this work constructs an offline dataset using heuristic migration trajectories and introduces a Decision Mamba-based policy network. Leveraging its capability for selective state-space sequence modeling, the framework captures resource contention and coupling among multiple VNFs. Combined with a twin-Critic architecture and a conservative regularization mechanism, our Offline RL framework jointly optimizes multiple objectives and provides a new solution for failure-driven multi-VNF parallel migration.

## 3. System Model and Problem Description

This section defines the mathematical model for failure-driven parallel VNF migration in edge networks, including the underlying network model, the modeling of Service Function Chains (SFCs) and Virtual Network Functions (VNFs), the migration scenario formulation, and the corresponding constraints and evaluation metrics.

### 3.1. Network Model

The substrate network is modeled as an undirected graph $G^{S}=(V^{S},E^{S})$, where $V^{S}$ denotes the set of physical nodes and $E^{S}$ denotes the set of physical links.

A physical node $v_{i}^{S}\in V^{S}$ represents the i-th physical node. Each physical node is equipped with multiple types of resources to support VNF execution. Its CPU and memory resources are denoted by $C_{i}^{S}$ and $M_{i}^{S}$, respectively, and the available residual resources are denoted by ${\tilde{C}}_{i}^{S}$ and ${\tilde{M}}_{i}^{S}$.

For the set of physical links $E^{S}$, an element $e_{pq}^{S}$ represents the physical link connecting neighboring nodes $v_{i}^{S}$ and $v_{j}^{S}$. The bandwidth and transmission delay of a physical link are denoted by $B_{pq}^{S}$ and $D_{pq}^{S}$, respectively, and the residual bandwidth is denoted by ${\tilde{B}}_{pq}^{S}$.

### 3.2. SFC and VNF

A SFC consists of multiple VNFs with different functionalities that are connected in a predefined business-logic order.

Let the set of SFCs be denoted by $G^{V}=\{G_{j}^{V}\}$, where the j-th SFC is modeled as a directed weighted graph $G_{j}^{V}=(V_{j}^{V},E_{j}^{V})$. Each SFC $G_{j}^{V}$ is associated with a traffic demand $F\left(G_{j}^{V}\right)$, a maximum tolerable downtime $T_{j}^{down}$, and a maximum tolerable delay $D_{j}^{V}$.

The virtual node set of the j-th SFC is denoted by $V_{j}^{V}=\{s_{j}^{V},v_{j1}^{V},\ldots ,v_{ju}^{V},\ldots ,d_{j}^{V}\}$, where $s_{j}^{V}$ and $d_{j}^{V}$ are the source and destination nodes, and $v_{ju}^{V}$ is the u-th VNF node. Let F denote the set of all VNF types, and $f\left(v_{ju}^{V}\right)\in F$ represent the specific VNF type of virtual node $v_{ju}^{V}$.

In the virtual link set $E_{j}^{V}$, an element $e_{ju}^{V}$ represents the virtual link connecting adjacent VNFs $v_{ju}^{V}$ and $v_{j\left(u+1\right)}^{V}$. To describe the mapping relationship between VNFs and physical nodes, we define a binary variable $x_{ju}^{i}$. When VNF $v_{ju}^{V}$ is deployed on physical node $v_{i}^{S}$, we set $x_{ju}^{i}=1$; otherwise, $x_{ju}^{i}=0$. Similarly, we define a binary variable $y_{ju}^{pq}$ to indicate the mapping of virtual links onto physical links. When virtual link $e_{ju}^{V}$ is mapped onto the physical link $e_{pq}^{S}$, we set $y_{ju}^{pq}=1$; otherwise, $y_{ju}^{pq}=0$.

### 3.3. SFC Migration

When a physical node fails, all VNFs deployed on that node must be migrated to ensure service continuity.

As shown in Figure 1, VNFs 1, 2, 3, and 4 of SFC1 are deployed on physical nodes 6, 2, 3, and 4, respectively. The red dashed line denotes the original service path of SFC1 before the occurrence of node failure.

Assume that physical node 3 fails, causing VNF3 of SFC1 to become unavailable and requiring its migration to another active node. In this scenario, VNF3 is migrated to server node 7, whose available CPU and memory resources meet the requirements of the migrating VNF. The green dashed line represents the re-mapped service path of SFC1 after migration.

Let the set of VNFs deployed on the faulty node $v_{f}^{S}\in V^{S}$ be defined as $V^{mig}=\{v_{ju}^{V}\in V^{V}|x_{ju}^{f}=1\}$, $V^{V}=\cup_{j}V_{j}^{V}$, where $V^{V}$ denotes the set of all VNFs for all SFCs.

For each affected SFC, we define: $G^{aff}=\{G_{j}^{V}\in G^{V}|\exists v_{ju}^{V}\in V_{j}^{V},v_{ju}^{V}\in V^{mig}\}$, i.e., all SFCs that contain at least one VNF to be migrated. Before selecting the target nodes for migration, the available resources of all physical nodes must be examined. Based on resource availability, link connectivity, and node health status, each migrating VNF selects a set of k nodes as candidate nodes to support subsequent service continuity. The candidate node set for each migrating VNF is defined as: $V_{ju}^{cand}\subseteq V^{S}$.

Considering that different types of VNFs consume different processing resources, we assume a linear relationship between resource consumption and traffic volume [16]. We define a resource demand coefficient: $coeff^{c}\left(f\left(v_{ju}^{V}\right)\right)$ and $coeff^{m}\left(f\left(v_{ju}^{V}\right)\right)$, where $coeff^{c}(\cdot )$ and $coeff^{m}(\cdot )$ denote the CPU and memory resources needed to process a unit of traffic for that VNF type.

Because different VNFs may increase or decrease the traffic volume during processing [17], we define the flow-scaling ratio of VNF type $f\left(v_{ju}^{V}\right)$ as: $ratio\left(f\left(v_{ju}^{V}\right)\right)$. Similarly, we define the memory-state dirty rate of VNF type $f\left(v_{ju}^{V}\right)$ as: $D\left(f\left(v_{ju}^{V}\right)\right)$.

For VNF $f\left(v_{ju}^{V}\right)$ in SFC $G_{j}^{V}$, its bandwidth demand is calculated as:(1)$$bw\left(e_{ju}^{V}\right)=F\left(G_{j}^{V}\right)\cdot \prod_{x=1}^{u}ratio\left(f\left(v_{jx}^{V}\right)\right)$$

Correspondingly, the CPU and memory demands of VNF $v_{ju}^{V}$ are defined as:(2)$$r_{ju}^{C}=coeff^{c}\left(f\left(v_{ju}^{V}\right)\right)\cdot bw\left(e_{j\left(u-1\right)}^{V}\right)$$(3)$$r_{ju}^{M}=coeff^{m}\left(f\left(v_{ju}^{V}\right)\right)\cdot bw\left(e_{j\left(u-1\right)}^{V}\right)$$

### 3.4. SFC Migration Constraints Under Failure Scenarios

To address the core requirements of migrating all VNFs from the faulty node, three key constraints are introduced.

#### 3.4.1. Complete Migration Constraint for VNFs on the Faulty Node

All VNFs deployed on the faulty node must be migrated, and each VNF can only be migrated to one feasible candidate node to ensure that no VNF is omitted. The constraint is formulated as:(4)$$\sum_{v_{i}^{S}\in V_{ju}^{cand}}x_{ju}^{i}=1,\;\forall v_{ju}^{V}\in V^{mig}$$

#### 3.4.2. Resource Constraints

The available resources of the target node must be sufficient to accommodate the total resource demands of all VNFs migrated to that node.

(a)CPU Resource Constraint:


(5)
$$\sum_{v_{ju}^{V}\in V^{mig}}\sum_{v_{i}^{S}\in V_{ju}^{cand}}x_{ju}^{i}\cdot r_{ju}^{C}\leq {\tilde{C}}_{i}^{S}$$


(b)Memory Resource Constraint:


(6)
$$\sum_{v_{ju}^{V}\in V^{mig}}\sum_{v_{i}^{S}\in V_{ju}^{cand}}x_{ju}^{i}\cdot r_{ju}^{M}\leq {\tilde{M}}_{i}^{S}$$


(c)Link Bandwidth Constraint

A physical link must support both service traffic and migration traffic. The total bandwidth demand must not exceed the link capacity:(7)$$\sum_{v_{ju}^{V}\in V^{mig}}y_{ju}^{pq}\cdot B_{ju}\leq {\tilde{B}}_{pq}^{S},\;\forall e_{pq}^{S}\in E^{S}$$

$B_{ju}$ denotes the migration bandwidth allocated to VNF $v_{ju}^{V}$ after determining the target node, which is computed by the PPLARBA bandwidth allocation algorithm. To ensure successful migration, the migration bandwidth must satisfy:$$B_{ju}\geq D\left(f\left(v_{ju}^{V}\right)\right),\;\forall v_{ju}^{V}\in V^{mig}$$

#### 3.4.3. End-to-End Delay Constraint

After migration, the end-to-end delay of each SFC must not exceed its maximum tolerable delay $D_{j}^{V}$. The constraint is formulated as follows:(8)$$\sum_{v_{ju}^{V}\in V_{j}^{V}}D_{proc}\left(f\left(v_{ju}^{V}\right)\right)+\sum_{e_{ju}^{V}\in E_{j}^{V}}\sum_{e_{pq}^{S}\in E^{S}}y_{ju}^{pq}\cdot D_{pq}^{S}\leq D_{j}^{V},\;\forall G_{j}^{V}\in G^{aff}$$
where, $D_{proc}\left(f\left(v_{ju}^{V}\right)\right)$ denotes the processing delay of the VNF of type $f\left(v_{ju}^{V}\right)$.

### 3.5. Evaluation Metrics for SFC Migration

In failure scenarios, Service Function Chain (SFC) migration is required to preserve service continuity while minimizing its impact on user experience and network resources. As such, maximizing the migration success rate stands as the core objective of SFC migration in fault-related contexts. Concurrently, it is also essential to keep both the migration time and migration cost incurred during the process within reasonable and controllable bounds.

#### 3.5.1. Migration Success Rate

According to the migration constraints defined earlier, if any constraint is violated during the migration process, the migration is considered unsuccessful. For an SFC to be deemed successfully migrated, all VNFs requiring migration within that SFC must be migrated successfully. Based on this criterion, the migration success rate is defined as:(9)$$\rho_{s}=\frac{N_{suc}}{\left|G^{aff}\right|}$$
where $N_{suc}$ denotes the number of successfully migrated SFCs, and $\left|G^{aff}\right|$ represents the total number of SFCs requiring migration.

#### 3.5.2. Migration Time

VNF migration adopts the pre-copy mechanism [18], in which the memory state of the VNF is continuously transferred during migration. When the allocated migration bandwidth B is greater than the dirty rate D, the amount of updated memory to be retransmitted decreases in each round, resulting in shorter transmission time per round, until the remaining data becomes sufficiently small and the process enters the final stop-and-copy phase. We define the ratio between the dirty rate and the allocated migration bandwidth as: $r=\frac{D\left(f\left(v_{ju}^{V}\right)\right)}{B_{ju}}$, when r < 1. After N rounds of pre-copy, the total migration time is:(10)$$t_{ju}^{mig}=\frac{V_{ju}}{B_{ju}}\times \frac{1-r^{N}}{1-r},\;V_{ju}=\lambda \cdot r_{ju}^{M}$$
where 0≤λ<1 denotes the fraction of memory pages updated during each iteration.

To control service downtime, the number of pre-copy rounds N must satisfy: $t_{down}=\frac{V_{ju}\cdot r^{N}}{B_{ju}}\leq T_{j}^{down}$, thus, $N=ceil\left(\log_{r}\left(T_{j}^{down}*\frac{B_{ju}}{V_{ju}}\right)\right)$, where ceil() denotes the ceiling function.

In the parallel migration scenario, the total migration time required for all VNFs is determined by the VNF with the longest migration duration:(11)$$T^{total}=\underset{v_{ju}^{V}\in V^{mig}}{\max}t_{ju}^{mig}$$

#### 3.5.3. Average Migration Cost

The migration cost reflects the overall consumption of network resources during the migration process as well as the impact caused by changes in service paths after migration. It primarily consists of two components: the resource consumption incurred during VNF migration and the rerouting overhead introduced by remapping service paths.

##### Migration Cost

During migration, VNFs generate network resource consumption that depends on the amount of state data to be transferred and the length of the migration path. The migration cost is defined as:(12)$$C^{mig}=\sum_{v_{ju}^{V}\in V^{mig}}V_{ju}\cdot h\left(v_{ju}^{V}\right)$$
where $h\left(v_{ju}^{V}\right)$ denotes the hop count of the migration path from the faulty node to the target node, obtained through the shortest-path computation in the PPLARBA algorithm.

##### Rerouting Cost

After migration, the virtual links of the affected SFCs must be remapped to new physical paths. These path changes alter network load and introduce additional overhead. The rerouting cost is defined as:(13)$$C^{rer}=\sum_{G_{j}^{V}\in G^{aff}}\sum_{e_{ju}^{V}\in E_{j}^{V}}\Delta h\left(e_{ju}^{V}\right)\cdot bw\left(e_{ju}^{V}\right)$$
where $\Delta h\left(e_{ju}^{V}\right)$ is the change in hop count of the virtual link before and after migration. If both endpoints of a virtual link are not migrated, then $\Delta h\left(e_{ju}^{V}\right)=0$, and the link incurs no rerouting cost.

##### Total Migration Cost

(14)$$C^{total}=\alpha \cdot C^{mig}+\beta \cdot C^{rer}$$
where α and β are weighting coefficients.

## 4. Algorithm Design

This paper proposes a Reinforcement Learning-Driven Parallel Migration Optimization (RL-PMO) method for Service Function Chain (SFC) migration in edge networks. The proposed approach targets the problem of parallel SFC migration under edge node failures and establishes a two-stage algorithmic framework that integrates heuristic-based offline data generation with offline reinforcement learning. The overall logical structure of the algorithm is illustrated in Figure 2.

In the first stage, an improved heuristic algorithm (CSSA or PSO) is employed to generate high-quality migration trajectories, addressing the common issues of low data quality and insufficient scenario coverage in offline datasets. Specifically, instead of retaining only the final optimized solutions, we also record all intermediate states and actions throughout the iterative optimization process, thereby constructing a rich and high-quality offline dataset with diverse samples.

In the second stage, the migration optimization process is formulated as a Markov Decision Process (MDP) and further transformed into a single-step sequential decision problem. Based on this formulation, we design a complete offline reinforcement learning framework composed of three core components:

Policy Network: The Decision Mamba model is adopted as the core decision module. Owing to its efficient temporal modeling capability, Mamba can effectively integrate state, goal, and temporal information from sequential inputs, enabling precise decision-making even in single-step migration scenarios [19,20].

Value Network: To stably evaluate the value of actions generated by the policy network and provide reliable optimization feedback, we employ a twin-Critic architecture.

These two identically structured but independently parameterized networks jointly estimate the value of each state–action pair. During policy gradient computation, the minimum value of the two Critics is used to mitigate overestimation, thereby improving the stability of training.

Conservative Learning: To address the distributional shift problem inherent in offline reinforcement learning [21], the Conservative Q-Learning (CQL) algorithm is incorporated into the Critic training objective. CQL encourages the Critics to assign higher values to actions within the data distribution while penalizing overly optimistic estimates of out-of-distribution actions, leading to a more conservative and robust value function.

Additionally, a PPLARBA module is designed to manage bandwidth scheduling and parallel execution of migration plans within the physical network, ensuring that the migration decisions generated by RL-PMO are practically executable in real deployment environments.

### 4.1. Offline Dataset Generation

During the trajectory construction phase, a hybrid heuristic method combining the Chaotic Sparrow Search Algorithm (CSSA) and the Particle Swarm Optimization (PSO) algorithm is employed. Traditional SSA algorithms are prone to premature convergence when handling complex optimization problems, and insufficient population diversity often weakens global search capability. To generate high-quality migration trajectories, this study enhances CSSA in both the initialization and update strategies.

In the initialization stage, a Tent chaotic mapping mechanism is introduced to improve population diversity. The mapping process is formulated as:(15)$$x_{i}^{0}=lb+\left(ub-lb\right)\cdot Z_{t},\;Z_{t+1}=\left\{\begin{array}{c} \frac{Z_{t}}{a},\;\;Z_{t}<a\\ \frac{1-Z_{t}}{1-a},\;\;Z_{t}\geq a \end{array}\right.$$
where $Z_{t}$ denotes the chaotic sequence generated by the Tent map, and $a\in \left(0,1\right)$ is the Tent mapping parameter that controls the distribution and complexity of the generated sequence.

In the updating phase, Gaussian perturbation and a symmetric reflection matrix are incorporated into the discoverer–joiner updating process to strengthen the balance between global exploration and local exploitation. The corresponding update rule is expressed as:(16)$$x_{i}^{t+1}=x_{best}^{t}+|x_{i}^{t}-x_{best}^{t}|\cdot {\left(AA\right)}^{T}$$
where A denotes a Gaussian random matrix. This enhancement enables the algorithm to achieve stronger global search capability and better local escaping ability, thereby improving the diversity and quality of the generated offline dataset.

In addition, CSSA introduces two enhancement mechanisms—Gaussian mutation for superior individuals and Tent disturbance for inferior ones—to further improve population diversity and convergence stability. The update strategy is defined as follows:(17)$$x_{i}^{t+1}=x_{i}^{t}\left(1+N\left(0,1\right)\right),x_{i}^{t+1}=x_{i}^{t}\left(1+Tent\left(t\right)\right)$$
where N(0,1) denotes a Gaussian random variable with zero mean and unit variance, and Tent(t) represents the chaotic perturbation generated by the Tent mapping.

For individuals with higher fitness, a slight Gaussian mutation is applied to enhance local exploration; for those with lower fitness, Tent disturbance is employed to prevent stagnation and premature convergence, thereby maintaining population diversity and enhancing global convergence robustness.

To avoid overfitting and limited data coverage that may result from using a single heuristic algorithm, PSO is additionally incorporated to generate trajectory data under various network conditions. This integration enriches the dataset and enhances the generalization capability of the Decision Mamba (DM) model during training.

Under the MDP framework, trajectory data are constructed in the canonical state–action–reward–next-state format:$$D=\left(s_{0},a_{0},r_{0},s_{1},\ldots ,s_{T},a_{T},r_{T}\right)$$

To cover service workloads of different scales, we randomly select 100–280 SFCs to be deployed in the physical network in each simulation round, thereby generating diverse network states. Based on these states, CSSA and PSO jointly produce a total of 831 migration trajectories, comprising 124,650 time steps for offline reinforcement learning training.

In addition, this study models SFC migration as a single-step parallel decision-making problem: given the current network state, the reinforcement learning agent selects target nodes for all VNFs in a single decision, without requiring sequential decisions or multi-step interactions.

### 4.2. Markov Decision Process (MDP) Modeling

In this work, we consider only single-node failure scenarios. At the beginning of each simulation episode, one physical node is selected uniformly at random from the node set and designated as the faulty node. Based on this selection, the corresponding set of VNFs to be migrated is constructed, and the initial state $s_{0}$ is generated.

The reinforcement learning process is formulated as a Markov Decision Process (MDP), denoted as $\langle S,A,P,R\rangle$, where each component is defined as follows.

State Space

The state space S is given by:(18)$$S=\left[v_{f}^{S},|V^{mig}|,{\left\{f_{i}^{VNF},r_{i}^{cand}\right\}}_{i}^{MAX_{mig}}\right]$$
where $v_{f}^{S}$ denotes the faulty node, $\left|V^{mig}\right|$ denotes the number of VNFs to be migrated. The migration-related features of the i-th VNF are defined as:(19)$$f_{i}^{VNF}=\{r_{ju_{i}}^{C},r_{ju_{i}}^{M},V_{ju_{i}},D\left(f\left(v_{ju_{i}}^{V}\right)\right),D_{j_{i}}^{V}\}$$
representing the CPU demand, memory demand, migration data volume, dirty rate, and maximum tolerable delay of the SFC to which the VNF belongs.

The migration-target-related features for candidate nodes are:(20)$$r_{i}^{cand}=\{{\tilde{C}}_{1}^{cand},{\tilde{M}}_{1}^{cand},\ldots ,{\tilde{C}}_{i}^{cand},{\tilde{M}}_{i}^{cand},\ldots ,{\tilde{C}}_{k}^{cand},{\tilde{M}}_{k}^{cand}\}$$
where $k=|V_{ju_{i}}^{cand}|$ is the number of candidate nodes for the i-th VNF. $MAX_{mig}$ is the upper bound on the number of VNFs that may require migration. Because the number of VNFs deployed on different physical nodes varies, when $|V^{mig}|<MAX_{mig}$ the remaining slots in the state vector are padded with zeros to ensure consistent state dimensionality.

Therefore, the total state dimension is: $2+\left(2k+5\right)MAX_{mig}$.

Action Space

(21)$$A=\{a|a_{1},a_{2},\ldots ,a_{i},\ldots ,a_{MAX_{mig}}\},\;a_{i}=\left[a_{i,1},\ldots ,a_{i,k}\right]$$
where $a_{i.j}$ denotes the probability weight of selecting the j-th candidate node for the i-th VNF. These weights satisfy: $\sum_{j=1}^{k}a_{i.j}=1$. which guarantees normalization of the decision policy.

State Transition Probability

The state transition probability $P\left(s\left(t+1\right)|s\left(t\right),a\left(t\right)\right)$ denotes the probability that, after executing action $a\left(t\right)$ in the current state $s\left(t\right)$, the system moves to the next state $s\left(t+1\right)$.

Reward Function

(22)$$R=\omega_{1}n_{t}\left(T^{total}\right)+\omega_{2}n_{s}\left(\rho_{s}\right)+\omega_{3}n_{c}\left(C^{total}\right)$$
where $\omega_{1},\omega_{2},\omega_{3}$ are the weights assigned to migration time, migration success rate, and migration cost, respectively. The terms $n_{t},n_{s},n_{c}$ denote the normalized migration time, success rate, and migration cost: $n_{t}\left(T^{total}\right)=\frac{{T^{total}}_{\max}-T^{total}}{{T^{total}}_{\max}-{T^{total}}_{\min}}$, $n_{s}\left(\rho_{s}\right)=\rho_{s}$, $n_{c}\left(C^{total}\right)=\frac{{C^{total}}_{\max}-C^{total}}{{C^{total}}_{\max}-{C^{total}}_{\max}}$, where ${T^{total}}_{\max}$, ${T^{total}}_{\min}$, ${C^{total}}_{\max}$ and ${C^{total}}_{\max}$ are the upper and lower bounds obtained from multiple rounds of preliminary simulations.

### 4.3. Offline Reinforcement Learning Policy

Given the trajectory dataset $D=\left(s_{0},a_{0},r_{0},s_{1},\ldots ,s_{T},a_{T},r_{T}\right)$, the training process converts it into an RTG-form sequence:$$D=(RTG_{0},s_{0},a_{0},RTG_{1},\ldots ,RTG_{K},s_{K},a_{K}),RTG_{t}=\sum_{k=t}^{T}\gamma^{k-t}r_{k}$$
where γ is the discount factor and K denotes the number of time steps, the parallel migration decision sequence covering K VNFs. For the single-step decision-making scenario, the RTG degenerates into the immediate reward corresponding to the current state–timestep pair: $RTG_{t}=r_{t}$.

DM serves as the core decision module: essentially, it learns a policy that generates high-quality migration plans conditioned on the current network state. It takes $D=(RTG_{0},s_{0},a_{0},RTG_{1},\ldots ,RTG_{K},s_{K},a_{K})$ as input and, through the Mamba encoder, captures the complex resource dependencies among VNFs and the long-horizon decision patterns, ultimately producing the action distribution $\pi_{\theta }\left(a_{t}|s_{t}\right)$.

The Critic loss consists of two parts: a Bellman regression term and a CQL conservative regularization term:(23)$$L_{Q}=L_{TD}+\lambda_{CQL}\cdot L_{CQL}$$

Here $L_{TD}$ is the Bellman regression loss, computed as(24)$$L_{TD}=\frac{1}{\left|B\right|}\sum_{(s,a,r)\in B}\left[{\left(Q_{\varphi_{1}}(s,a)-y\right)}^{2}+{\left(Q_{\varphi_{2}}(s,a)-y\right)}^{2}\right]$$

In the above, B denotes a mini-batch sampled from the dataset, and y is the target value. For the single-step decision setting, we take y=r to avoid training oscillation.

The conservative regularization term $L_{CQL}$ is defined as(25)$$L_{CQL}=\frac{1}{\left|B\right|}\sum_{s\in B}\left\{[log(\frac{1}{K}\sum_{k=1}^{K}exp\left(Q_{\varphi_{1}}\left(s,{\tilde{a}}_{k}\right)\right)-Q_{\varphi_{1}}\left(s,a\right)]+[log(\frac{1}{K}\sum_{k=1}^{K}exp\left(Q_{\varphi_{2}}\left(s,{\tilde{a}}_{k}\right)\right)-Q_{\varphi_{2}}\left(s,a\right)]\right\}$$

Here, ${\tilde{a}}_{k}$ denotes actions sampled from the policy network, i.e., ${\tilde{a}}_{k}\sim \pi_{\varphi }\left(\cdot |s\right),\;k=1,\ldots ,K$.

During training, the main Q-network parameters are updated jointly with the soft target update.

Because the state space and modeling structure of Mamba naturally align with the multi-VNF parallel migration problem studied in this work—where complex resource-competition patterns exist among VNFs—the decision Mamba is adopted as the policy network. Mamba can capture sequential features and implicitly model global and local resource dependencies over time. Compared with transformers, Mamba is more suited for processing long sequences and optimization-related tasks.

The actor aims to maximize the value of the action estimated by the critics. Its loss is(26)$$L_{\pi }(\theta )=-\frac{1}{\left|B\right|}\sum_{s\in B}\min(Q_{\varphi_{1}}(s,a_{\pi }(s)),\;Q_{\varphi_{2}}(s,a_{\pi }(s)))$$
where θ are the DM model parameters and $a_{\pi }(s)$ is the action generated by the actor.

The actor parameters are updated by gradient descent:(27)$$\theta^{\prime }=\theta -\eta_{learning}\;\nabla_{\theta }L_{\pi }$$

These objectives guide the offline RL agent to produce reliable SFC migration decisions in NFV environments by discouraging over-optimistic Q-values while improving the policy that selects target nodes for VNF migration.

Target-network soft update. We adopt a soft-update rule to update the Critic target networks:(28)$$\varphi_{i}^{-}=\tau \cdot \varphi_{i}+(1-\tau )\cdot \varphi_{i}^{-}\;\;i=1,2$$

To prevent the policy network from prematurely tracking a Q-function that has not yet converged, we introduce an update-frequency parameter d: during training, the actor (policy network) is updated only once after every d critic (Q-network) updates, resulting in an update ratio of d:1 between the critic and the actor.

Execution layer. After obtaining the migration candidates for all VNFs, the RL agent interacts with the NFV environment to generate effective VNF migration decisions. Once the actor outputs target nodes, the PPLARBA algorithm is invoked to allocate migration bandwidth for each VNF and evaluate migration feasibility, thereby completing one full migration cycle.

In summary, the pseudocode of our RL-PMO algorithm is given in Algorithm 1:


**Algorithm 1:** Reinforcement Learning-Driven Parallel Migration Optimization AlgorithmInputs: Offline data offline dataset D; number of epochs E; number of training steps T; soft-update coefficient τ; CQL factor $\lambda_{CQL}$; batch size B; update frequency d; learning rate $\eta_{learning}$Output: SFC migration policy $\pi_{\theta }$Initialize Critic1, Critic2 and Actor (DM) parameters as $\varphi_{1},\varphi_{2},\theta$
Initialize target Critic parameters $\varphi_{1}^{-}\leftarrow \varphi_{1}$, $\varphi_{2}^{-}\leftarrow \varphi_{2}$1for epoch = 1 to E do2  for i = 1 to T do3      Sample a mini-batch $B=\{s_{t},a_{t},r_{t},\ldots ,s_{t+B},a_{t+B},r_{t+B}\}$ from D4    with no_grad:5      Generates policy actions ${\tilde{a}}_{k}$.6    The main Critic networks predict Q-values: Q1, Q2.7    Calculate Critic loss $L_{Q}$ using Equations (23)–(25)8    Update main Critic parameters $\varphi_{1},\varphi_{2}$ by gradient descent.9    if i % d == 0 then10      Generate the policy action in the current state: $a_{\pi }=\pi_{\theta }(\cdot |s)$11      Calculate actor loss $L_{\pi }(\theta )$ by Equation (26)12      Update actor parameters θ by Equation (24)13    end if14       Soft-update target critics by Equation (28): $\varphi_{i}^{-}\leftarrow \tau \varphi_{i}+(1-\tau )\varphi_{i}^{-}$15  end for16end for17return $\pi_{\theta }$


## 5. Performance Evaluation

### 5.1. Simulation Setup

We use the open-source NFV resource-allocation simulator SFCSim [22]. The reinforcement learning models are implemented using PyTorch 2.2.1 and executed on a Linux-based system. The substrate network is a cellular topology with 127 nodes and 342 links; each node offers 20 CPU cores and 64 GB of memory. Each physical link has a fixed bandwidth of 8 Gbps and a latency of 0.5 ms.

The catalog includes 8 VNF types. For each VNF, the traffic scaling factor follows U(0.8, 1.2); the CPU coefficient per MBps lies in [0.05, 0.10], and the memory coefficient lies in [80, 100] MB/MBps. The dirty-page rate is fixed at 50 MBps. Service demand follows U(10, 20) MBps, and the SFC length (number of VNFs per chain) ranges from 3 to 6.

#### Network Architecture and Hyperparameter Configuration

For the heuristic baseline algorithms, the population size of the Chaotic Sparrow Search Algorithm (CSSA) is set to 30, with 150 iterations, and the Tent chaotic mapping parameter is configured as a=0.7. For the Particle Swarm Optimization (PSO) algorithm, the swarm size is set to 30, with 100 iterations.

The Decision Mamba policy network consists of four Mamba blocks with a hidden dimension of 256. A final linear output layer is used to generate the probability distribution over candidate nodes. Each of the twin-critic value networks contains three fully connected hidden layers with 256 neurons per layer.

Both the policy network and the critic networks are optimized using the Adam optimizer, with a learning rate of $3\times 10^{-4}$. The key reinforcement learning hyperparameters include a discount factor of γ=0.99, a soft update coefficient of τ=0.005, and a CQL regularization weight of $\lambda_{CQL}=0.5$.

### 5.2. Simulation Result and Analysis

We assess each method under varying SFC counts using three metrics: migration success rate, migration time, and migration cost. Figure 3 report the results for six SFC-migration algorithms across different load levels; the x-axis is the number of deployed SFCs, which serves as the load indicator.

From Figure 3a, BC (behavior cloning) merely reproduces trajectories present in the dataset; it is highly dependent on data quality and generalizes poorly to unseen fault scenarios, leading to weaker performance. DT (Decision Transformer) is relatively stable, with a migration success rate of roughly 80%. By contrast, DM (Decision Mamba) extracts richer contextual and structural cues from the inputs and therefore achieves a higher overall success rate than DT. The standard IQL algorithm (without fine-tuning) exhibits moderate performance [23], achieving 87% under a load of 140 SFCs but dropping to 77% when the load increases to 280 SFCs. IQL+FT, which incorporates online fine-tuning, shows more stable performance compared with plain IQL; however, its effectiveness still degrades noticeably under heavier loads, falling to around 85% at 280 SFCs. This indicates that although brief online fine-tuning is beneficial, its generalization capability remains limited under high-load conditions with severe resource contention. We also observe that as the network load increases, the success rates of BC and DM drop noticeably, indicating that intensified resource contention makes migration decisions more constrained by underlying compute and bandwidth limits. Notably, RL-PMO, which augments DM with twin-critic value estimation, sustains a success rate above 90% and degrades more slowly as load grows, demonstrating stronger robustness and reliability under high-load stress.

From Figure 3b, as load (number of SFCs) increases, the migration time of all six algorithms rises. Overall, BC, DT, and DM exhibit similar average migration time across load levels, with differences of about 0.2 s. The overall migration time of the IQL algorithm is relatively stable and remains at a low level. Under light load, RL-PMO is slightly faster than the other five; however, as the load grows, its average migration time increases more noticeably, and when the SFC count exceeds 200, RL-PMO becomes slower than the baselines. This behavior stems from RL-PMO’s design choice to favor more robust but slightly slower migration and bandwidth-allocation plans under tight resources in order to preserve a high success rate. The resulting overhead is expected and acceptable, and in a proactive migration setting—where failures are predicted and migration is triggered in advance—such fraction-of-a-second increases have negligible impact on service continuity.

Figure 3c compares the migration cost of all methods. BC is consistently and significantly more expensive than the others, indicating that pure imitation learning struggles to make resource-efficient decisions in complex, dynamic NFV/SFC environments. Consistent with Figure 3b, RL-PMO performs better under light load; however, as the load grows, it favors more resource-rich target nodes and conservative bandwidth allocation to keep the success rate stable, leading to a modest increase in both cost and time. In fault-driven scenarios, maintaining a high and stable success rate is paramount, so a controlled increase in time and cost is a reasonable trade-off.

To provide a comprehensive quantitative comparison, Table 1 summarizes the average performance of all algorithms across varying load levels (140–280 SFCs). 

As shown in Table 1, RL-PMO achieves an average migration success rate of 95.26% across all load levels, outperforming IQL+FT and DM by approximately 9.2 and 9.7 percentage points, respectively, demonstrating significant superiority over other offline reinforcement learning baseline methods. BC exhibits an average success rate of only 74.73%, indicating that pure behavioral cloning fails to maintain reliable migration performance across diverse load scenarios. Although IQL achieves slightly better average migration time (3.02 s) and migration cost (21.94), its success rate is only 84.09%. In fault-driven SFC migration scenarios, success rate takes priority over time and cost. Overall, RL-PMO maintains a high success rate while controlling migration time at approximately 3.09 s and migration cost at around 24.41, achieving a reasonable trade-off between reliability and efficiency in multi-VNF parallel migration scenarios.

We further analyze the performance of RL-PMO under different values of k. In SFC migration, determining how many candidate nodes to assign to each VNF is a key trade-off: a larger candidate set expands the search space and may lead to better solutions, but it also increases decision complexity and computational overhead. To evaluate this trade-off, we set k = 4, 6, 8 and 10, compare the migration performance of RL-PMO under each configuration, and examine its sensitivity across different network load conditions.

As shown in Figure 4a, the success rate remains above 85% under all load levels, but clear performance differences appear across different values of k. Under light load, k = 4 achieves a success rate of 93.5%, while k = 6, k = 8 and k = 10 reach 97.2%, 97.0%, and 98.7%, respectively. Even in such resource-rich conditions, using more candidate nodes still yields a 3–5% improvement.

As the load increases, the success rate of k = 4 drops noticeably, whereas k = 6, k = 8 and k = 10 show better stability. Among them, k = 10 maintains a success rate between 92% and 98.7% across all load levels, and still reaches 92.2% at the highest load (280 SFCs), outperforming k = 4 by about 2.4 percentage points. The performance of k = 8 is similarly strong.

As shown in Figure 4b, migration time increases as the number of SFCs grows, but the differences across k values remain mild and follow a clear pattern. Under light load (140 SFCs), the migration times for all four k settings are almost identical, ranging from 2.5 to 2.6 s. This indicates that when resources are sufficient, the number of candidate nodes has minimal influence on migration time. As the load continues to rise, the time differences gradually appear, but under heavy load the curves for all four k values converge. This shows that under high load, the true bottleneck is the limitation of physical resources such as link bandwidth, rather than the number of candidate nodes.

As shown in Figure 4c, under light load, the migration cost for all four k values is concentrated within 16–17.5 units, with only small differences. However, as the load increases, the cost gap among different k values becomes more pronounced. Moreover, as the network load grows, the migration cost exhibits a similar pattern to the migration time, first increasing and then gradually converging. These results indicate that the choice of k does have an impact on performance optimization: overly simplifying the action space is not always beneficial, and appropriately enlarging the action space can help the algorithm find higher-quality solutions.

We then conducted multiple tests of RL-PMO on a 19-node cellular network topology. It is important to emphasize that all training data for RL-PMO were collected exclusively from the 127-node topology, and the algorithm had never encountered any information from the 19-node network during training. This experiment is designed to verify whether the algorithm has learned transferable underlying strategies, rather than merely memorizing superficial characteristics of a specific topology.

In this design, the 19-node network was configured with node resources and link parameters similar to those of the 127-node topology, while its overall size was reduced by approximately 85%. To match the reduced network scale, the number of SFCs used for testing was adjusted to 30–100. As shown in Figure 5a, the success rate fluctuates between 76.3% and 93.2%, exhibiting larger variance compared with the experiments on the 127-node topology. As shown in Figure 5b,c, both the migration time and migration cost also increase noticeably as the network load grows. However, such increases are reasonable. More importantly, the algorithm does not experience cost escalation or collapse when exposed to an unseen topology. This demonstrates that RL-PMO has captured the underlying logic of SFC migration rather than memorizing a specific environment. Even without having seen the 19-node topology during training, the algorithm is still able to complete 76–94% of the migration tasks, providing strong evidence that the learned policy possesses cross-topology transferability and environmental adaptability.

Finally, we evaluated the performance of the proposed RL-PMO framework under different neural network depths to examine its sensitivity to model complexity. As the number of hidden layers increases, the total number of neurons grows correspondingly, leading to higher computational overhead during both training and inference. Figure 6 illustrate the performance of RL-PMO with three, four, and five hidden layers under various network load levels. The results indicate that the number of hidden layers has only a marginal effect on RL-PMO’s overall performance. Across all configurations, RL-PMO consistently achieves higher migration success rates than the DT, DM, IQL, BC and IQL+FT algorithms. Under light-load conditions, RL-PMO also attains lower migration time and cost; however, as the network load increases, its migration time and cost become slightly higher than those of the baseline algorithms. This trade-off reflects RL-PMO’s conservative resource allocation strategy designed to maintain high reliability under heavy-load conditions.

Figure 7 and Figure 8 further depict the training dynamics of Decision Mamba and RL-PMO with different hidden-layer depths. For Decision Mamba (Figure 7), the model with three hidden layers converges to a lower average return compared with those with four or five layers, stabilizing after roughly 20 iterations. In contrast, RL-PMO (Figure 8) shows minimal sensitivity to network depth: its training curves for all configurations converge smoothly to an average return of around −110 after about 20 iterations. These observations demonstrate that RL-PMO not only achieves more stable and better-quality convergence than DM but also maintains robustness and adaptability across different neural network depths.

## 6. Conclusions

This study investigates the optimization of Service Function Chain (SFC) migration in Network Function Virtualization (NFV) environments under impending node failure scenarios and proposes a Reinforcement Learning-driven Parallel Migration Optimization algorithm (RL-PMO). On the decision side, RL-PMO leverages the Decision Mamba model for sequential modeling and contextual representation, integrating high-quality offline migration trajectories generated by heuristic algorithms such as CSSA and PSO to enable efficient high-dimensional state encoding and policy learning in complex migration environments. On the evaluation side, a twin-critic structure coupled with Conservative Q-Learning (CQL) regularization is employed to penalize out-of-distribution actions, thereby enhancing policy stability and generalization under heavy-load conditions.

Extensive simulation results show that RL-PMO maintains nearly a 95% migration success rate across different scales and load conditions, and its performance degrades slowly as network load increases. In terms of migration efficiency and cost, RL-PMO performs better under low load, while under high load, it proactively selects more reliable paths and resource allocations to ensure successful migration. Overall, RL-PMO achieves higher success rates and more stable performance with only small additional overhead, demonstrating strong robustness in complex NFV environments. This capability is particularly important in sensor-network and IoT-edge systems that rely on stable, continuous, and low-latency SFCs to maintain the reliability and responsiveness of critical applications.

Although RL-PMO demonstrates strong performance in the simulation environment, several directions remain worth further exploration: The current offline dataset is constructed from only a single 127-node topology. Future work can incorporate multiple network topologies during training to improve cross-topology generalization. This study focuses on batch VNF migration triggered by a single-node failure, and future research may extend this to multi-node failure scenarios and coordinate parallel migration across multiple failure regions. The current evaluation covers network sizes of up to 127 nodes. Future work could further scale the study to networks with thousands of nodes and explore model compression, knowledge distillation, and distributed inference to reduce computational overhead.

## Figures and Tables

**Figure 1 sensors-26-00242-f001:**
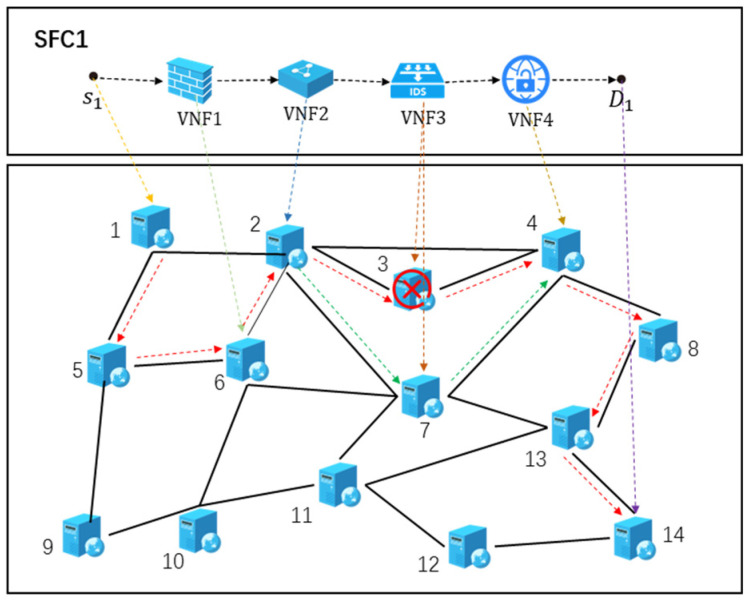
The migration of SFCs.

**Figure 2 sensors-26-00242-f002:**
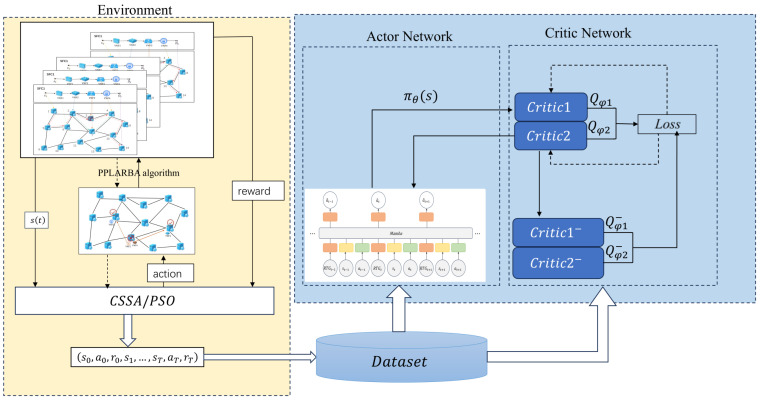
Overall framework of the RL-PMO.

**Figure 3 sensors-26-00242-f003:**
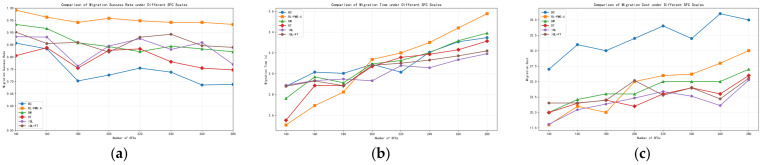
Comparison of SFC migration performance of different algorithms under different network load conditions: (**a**) migration success rate; (**b**) migration time; (**c**) migration cost.

**Figure 4 sensors-26-00242-f004:**
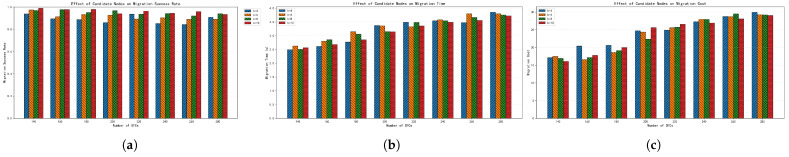
Impact of different numbers of candidate nodes on algorithm performance: (**a**) migration success rate; (**b**) migration time; (**c**) migration cost.

**Figure 5 sensors-26-00242-f005:**
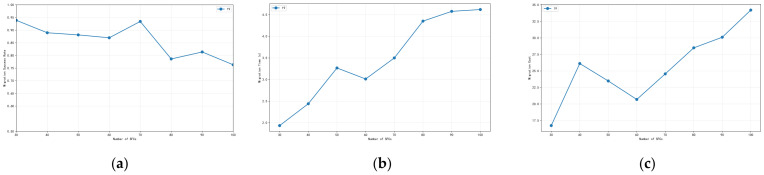
Performance of RL-PMO algorithm under different topologies: (**a**) migration success rate; (**b**) migration time; (**c**) migration cost.

**Figure 6 sensors-26-00242-f006:**
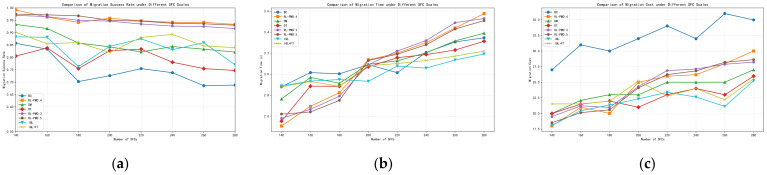
Comparison of RL-PMO algorithms with different numbers of hidden layers: (**a**) migration success rate; (**b**) migration time; (**c**) migration cost.

**Figure 7 sensors-26-00242-f007:**
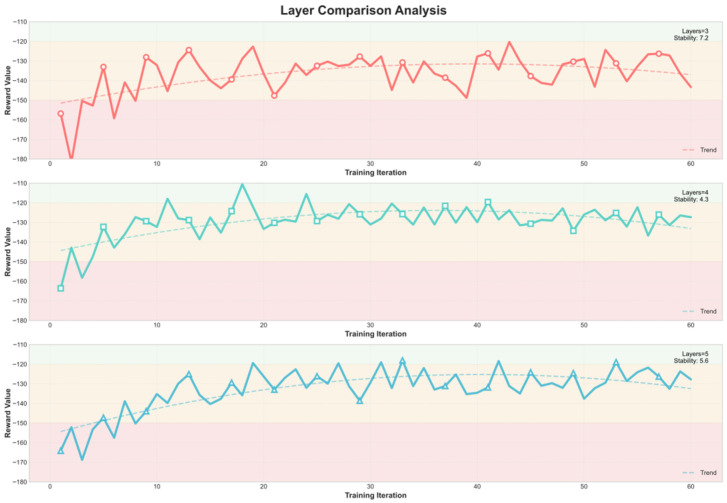
Training process of the DM algorithm under different hidden-layer depths.

**Figure 8 sensors-26-00242-f008:**
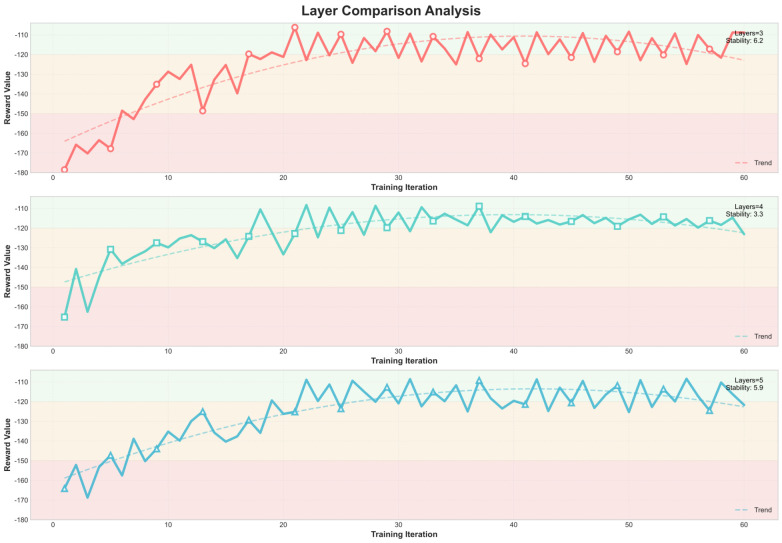
Training process of the RL-PMO algorithm under different hidden-layer depths.

**Table 1 sensors-26-00242-t001:** Average optimization target performance of the algorithm under different network load conditions.

Algorithm	Migration Success Rate	Migration Time	Migration Cost
RL-PMO	0.95257	3.09393	24.41
DM	0.85597	3.0985	23.7936
DT	0.7969	3.0436	22.4975
BC	0.74727	3.1017	32.2223
IQL	0.8409	3.0188	21.94397
IQL+FT	0.86083	3.05413	23.19163

## Data Availability

The original contributions presented in this study are included in the article. Further inquiries can be directed to the corresponding author.

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
