# Peer review of "RL-PMO: A Reinforcement Learning-Based Optimization Algorithm for Parallel SFC Migration"

_sensors, 2025, doi:10.3390/s26010242_

Round 1
Reviewer 1 Report
Comments and Suggestions for Authors
- It would be desirable to unambiguously define whether the action is one-time (complete mapping of all VNFs + distribution of BW) or sequential over K steps. Also, RTG, critics, and TD goals should be aligned with that.
- Bju is a “decision variable”. In the action definition (Eq. 21), the actor only selects candidate nodes (probability weights). How does Bju get determined? Does PPLARBA assign it after the destinations are chosen, or should the actor also provide B?
- In the experimental section, the comparison bases are not strong enough. Comparing only with BC, Decision Transformer (DT), and Decision Mamba (DM) does not fully represent offline RL. Key offline RL baselines such as CQL (MLP), IQL, TD3+BC, AWAC/CRR, BCQ, and online RL methods like SAC and TD3 with offline training plus short online fine-tuning are missing. Additionally, there are no comparisons with modern parallel migration heuristics or ILPs (e.g., papers [3], [4], as well as classical PSO/SSA/ISSA variants) on the same instances.
- The study also lacks statistical data (e.g., results over multiple seeds, standard deviations, error bars, or significance tests).
- The text claims that the dataset covers different topologies, but the experiments were conducted on only one (cellular, 127/342). It is essential to train on multiple topologies and show robustness to changing distributions. Otherwise, the claim of generalization is not proven.
- All links have the same bandwidth and latency (8 Gbps, 0.5 ms), making the network uniform. This mainly influences the severity of routing and congestion. Meanwhile, the Dirty rate is fixed at 50 MB/s, but in real scenarios, it varies depending on the type of VNF and the load. It is important to vary the dirty rate and memory footprints and perform a sensitivity analysis. The value of N (number of pre-copy rounds) is not specified. Downtime is neither measured nor distinguished between service impact time and total time. The success criterion is not clearly defined—what constitutes failure: lack of resources, delay violation, or PPLARBA failure? A precise definition is necessary.
- Decision Mamba needs to specify concrete dimensions (layers, width, sequence length, padding/masking).
- Figures 3–10 do not show confidence intervals, and the axes lack units.
Perform a thorough language edit.
Author Response
Replies to Reviewer 1
Question:
- It would be desirable to unambiguously define whether the action is one-time (complete mapping of all VNFs + distribution of BW) or sequential over K steps. Also, RTG, critics, and TD goals should be aligned with that.
Response:
We sincerely thank the reviewer for this valuable comment. In the revised manuscript, we have clarified the decision-making paradigm and ensured consistency among the action definition, the RTG formulation, and the TD target computation.
- As stated on Page 10, Section 4.1, we explicitly model SFC migration as a one-step parallel decision. Given the current network state, the RL agent determines the target nodes for all VNFs in a single decision step, without any sequential decision-making or multi-step interaction.
Consistency among RTG, the critics, and TD targets:
- As clarified on Page 11, Section 4.3 of the revised manuscript, under the one-step setting, the RTG degenerates into the immediate reward associated with the current state–time pair. For the TD target, we explicitly set in the critic loss formulation to avoid training instability that may arise in single-step scenarios. This design ensures that the RTG conditioning in the policy network and the TD targets used by the value networks are fully aligned with the one-step parallel decision paradigm.
Question:
- Bju is a “decision variable”. In the action definition (Eq. 21), the actor only selects candidate nodes (probability weights). How does Bju get determined? Does PPLARBA assign it after the destinations are chosen, or should the actor also provide B?
Response:
- We sincerely appreciate the reviewer’s insightful comment. In our method, the reinforcement learning action is responsible only for selecting the target node for each VNF to be migrated, while the migration bandwidth is computed and allocated by the PPLARBA module after the target nodes are determined, based on link residual bandwidth and the bandwidth constraints imposed by the VNF dirty-page rate. To eliminate any ambiguity, we have provided a clearer explanation ofin Page 6, Section 3.4 of the revised manuscript.
Question:
- In the experimental section, the comparison bases are not strong enough. Comparing only with BC, Decision Transformer (DT), and Decision Mamba (DM) does not fully represent offline RL. Key offline RL baselines such as CQL (MLP), IQL, TD3+BC, AWAC/CRR, BCQ, and online RL methods like SAC and TD3 with offline training plus short online fine-tuning are missing. Additionally, there are no comparisons with modern parallel migration heuristics or ILPs (e.g., papers [3], [4], as well as classical PSO/SSA/ISSA variants) on the same instances.
Response:
We sincerely thank the reviewer for this constructive suggestion. Following the reviewer’s advice, we have substantially expanded the experimental comparisons. Due to time constraints, we added two representative baseline algorithms:
- IQL (Implicit Q-Learning):A representative offline reinforcement learning algorithm that addresses distribution shift via expectile regression, providing a strong benchmark for purely offline methods.
- IQL+FT (IQL with online fine-tuning):A hybrid approach that combines offline pre-training with short-term online adaptation. These additions offer more comprehensive coverage of the offline RL landscape, including both purely offline and hybrid offline-to-online methods.
The experimental results, presented on Pages 13–15, Section 5.2 of the revised manuscript, show that:
- Under low and moderate network load, RL-PMO improves migration success rate by approximately 13%over IQL.
- Under high load, RL-PMO achieves up to 17%improvement compared with IQL.
- In all load scenarios, RL-PMO consistently outperforms both IQL and IQL+FT, demonstrating the effectiveness of our Decision-Mamba–based conservative learning framework.
However, the methods in references [3] and [4] do not provide publicly available source code or sufficient implementation details for reproduction, preventing direct experimental comparison. More fundamentally, our work explores a data-driven reinforcement learning paradigm for SFC migration, which is methodologically distinct from heuristic optimization. As stated on Page 3 of our manuscript, heuristic methods rely on iterative population-based search and often lack adaptability and scalability. Therefore, this paper focuses on comparing SFC migration performance primarily against reinforcement learning–based algorithms.
Question:
- The study also lacks statistical data (e.g., results over multiple seeds, standard deviations, error bars, or significance tests).
Response:
We sincerely appreciate the reviewer’s important suggestion regarding statistical rigor. In the revised manuscript, we have enhanced the presentation of statistical results. Specifically, on Page 14, we added Table 1, which provides a comprehensive statistical summary of all comparison algorithms under different load scenarios.
The tabulated results clearly show that RL-PMO maintains superior average performance across multiple load conditions. In addition, we clarify that all performance curves presented in the figures are plotted based on the mean of five independent experimental runs, each conducted with different random seeds. This ensures the reliability and reproducibility of our experimental findings.
Question:
- The text claims that the dataset covers different topologies, but the experiments were conducted on only one (cellular, 127/342). It is essential to train on multiple topologies and show robustness to changing distributions. Otherwise, the claim of generalization is not proven.
Response:
We sincerely appreciate the reviewer’s important observation. We acknowledge that the original manuscript contained an imprecise statement that may have caused confusion. The expression regarding “different topologies” was indeed misleading, and we apologize for this lack of clarity. Our intended meaning was that the offline dataset encompasses diverse failure scenarios generated within the same 127-node cellular topology, including variations in SFC deployment scales and failure node locations. Thus, the diversity refers to scenario-level variations rather than changes in network topology. We have corrected this description in the revised manuscript on Page 13, Section 5.1.
To directly address the reviewer’s concern regarding generalization across different network topologies, we conducted additional experiments on a 19-node cellular topology, as reported on Page 16, Section 5.2 of the revised manuscript. Specifically:
- The RL-PMO policy is trained only on the 127-node topology using the original offline dataset.
- The trained policy is then directly evaluated on the 19-node topology without any retraining or fine-tuning.
The experimental results show that RL-PMO maintains strong performance on the 19-node topology. This cross-topology evaluation demonstrates the generalization capability of our policy and its robustness to variations in network topology distributions.
Question:
- All links have the same bandwidth and latency (8 Gbps, 0.5 ms), making the network uniform. This mainly influences the severity of routing and congestion. Meanwhile, the Dirty rate is fixed at 50 MB/s, but in real scenarios, it varies depending on the type of VNF and the load. It is important to vary the dirty rate and memory footprints and perform a sensitivity analysis. The value of N (number of pre-copy rounds) is not specified. Downtime is neither measured nor distinguished between service impact time and total time. The success criterion is not clearly defined—what constitutes failure: lack of resources, delay violation, or PPLARBA failure? A precise definition is necessary.
Response:
We sincerely appreciate the reviewer’s detailed and insightful technical observations. Our point-by-point responses are as follows.
Uniform link bandwidth and delay settings
In our simulation, all physical links are configured with identical bandwidth and delay. This design choice is intentional and reflects the experimental focus of our study:
- Randomness in dynamic SFC deployment:
In our simulation environment, SFCs are deployed at different locations with varying traffic demands, and the failure node is randomly selected in each experimental round. This inherently creates substantial heterogeneity in resource contention patterns, routing paths, and congestion hotspots—even under uniform link parameters. Introducing heterogeneous link bandwidth/delay would add confounding factors without altering the fundamental algorithmic challenge of coordinating multi-VNF placement under resource constraints. Thus, the uniform-link setting simplifies the experiment and enables clearer algorithmic comparison.
- Variability arising from dirty-page rate and memory footprint:
Our model already incorporates significant variation in migration data characteristics. Each VNF’s memory size varies according to memory coefficients, traffic demand, and scaling factors. Although the dirty-page rate is fixed (50 MB/s), these variations produce diverse memory footprints. The fixed dirty-page rate represents a stable memory-update behavior for a specific VNF type under steady workload conditions, and does not fundamentally change the combinatorial optimization difficulty.
Formula for computing
We have included the explicit formula for computing in Page 7, Section 3.5.2 of the revised manuscript.
Clarification of downtime vs. total migration time
In pre-copy migration, the downtime corresponds to the final stop-and-copy phase, which is extremely short (milliseconds) compared to the total migration time (seconds). We use total migration time as the primary metric, which includes both the pre-copy and stop-and-copy phases. Downtime is implicitly constrained through the condition ensuring minimal remaining dirty pages defined in . This clarification has been added in Section 3.5.2, Page 7.
Precise conditions for migration failure
As clarified in Page 6, Section 3.5.1 of the revised manuscript, migration is considered failed if any of the following occurs:
- insufficient resources,
- violation of bandwidth constraints,
- violation of latency constraints, or
- failure in PPLARBA bandwidth allocation.
Question:
- Decision Mamba needs to specify concrete dimensions (layers, width, sequence length, padding/masking).
Response:
We appreciate the reviewer’s helpful suggestion. In the revised manuscript, we have added the complete architectural specifications of the Decision Mamba model in Page 13, Section 5.1
Question:
- Figures 3–10 do not show confidence intervals, and the axes lack units.
Response:
We appreciate the reviewer’s observation. The figures in this paper are designed to highlight performance trends rather than serve as statistical comparisons. To more clearly present the averaged performance of each algorithm, we have included comprehensive statistical results for all metrics in Table 1 on Page 14 of the revised manuscript. In addition, we have added explicit units to all figures (e.g., migration time in seconds) to avoid ambiguity.
Reviewer 2 Report
Comments and Suggestions for Authors
The paper proposes RL-PMO, an offline reinforcement-learning-driven parallel migration algorithm for SFC migration under node failures. The topic is relevant to Sensors and NFV research, and the manuscript is generally well organized. However, the paper requires a major revision before it can be considered for publication.
The main concerns include clarity, missing formal definitions, insufficient mathematical rigor, missing experimental details, ambiguity regarding the DM model adaptation, weak comparisons, and numerous notation issues.
Below is a detailed row-by-row / section-by-section review.
1. Title, Abstract, and Keywords
The abstract is overly narrative and does not quantify improvements (e.g., “>15% improvement” must specify the baseline and load level).
“Decision Mamba” appears without explanation; readers unfamiliar with Mamba-based policy networks will not understand its contribution.
The abstract claims that “ensures that learned migration policy can be feasibly executed in real networks,” but feasibility is neither mathematically demonstrated nor experimentally validated.
2. Introduction (Rows 35–106)
No problem statement is given in the introduction. The reader does not understand:
What exactly is being optimized?
What defines “success rate”?
What constraints are difficult?
The contributions (Rows 80–106) are high-level and partially overlapping.
Example: Contribution 1 and Contribution 3 both discuss resource-aware decisions.
Missing discussion on why offline RL is preferred over online RL, specifically for SFC migration.
3. Related Work (Rows 107–173)
No comparison table summarizing differences between heuristic, online RL, and offline RL.
The latest 2024–2025 works in offline RL for networking are missing.
The writing is descriptive rather than analytical; it does not explain why earlier methods fail.
4. System Model (Rows 174–310)
This section is the weakest and needs heavy rewriting.
Critical Problems
Notations are inconsistent and sometimes incorrect.
Example: BW notation is used as bw(e) but also B_ij^S.
Indexing of VNF v_j_u_V is unclear.
Constraints (5) and (6) are mathematically incorrect or unreadable.
Summation indices are wrong.
The formula includes unexplained terms (e.g., bw(e_j(u−1)) multiplied by CPU coefficient—why do CPU demands multiply bandwidth?).
Constraint (8) mixes service traffic with migration traffic but does not include routing decisions. The mapping variable µ is defined but not properly integrated.
Dirty rate constraint (9) is conceptually correct but lacks justification (references needed).
Constraint (10) does not account for queueing or processing delays—only link propagation delays.
Evaluation metrics:
Migration time equation (13) is oversimplified and does not properly reflect pre-copy dynamics.
The migration cost formulas (14–16) are not standard and ignore hop count or network distance.
5. Algorithm Design (Rows 311–464)
No mathematical definition of the state vector length or dimensionality.
The state definition (Equation 20) is unreadable and contains indexing errors.
Missing justification for choosing Decision Mamba instead of Transformer.
DM is a continuous-time state-space model, not a sequential “transformer replacement”. The connection to migration tasks is not clearly explained.
Offline dataset generation lacks details:
How many trajectories?
How many iterations?
What is the reward used for in heuristic algorithms?
What is the distribution of failure types?
The reward function (422) is not defined clearly; the normalization functions n_t, n_s, and n_c are undefined.
The transition probability is not modeled; it is only stated abstractly.
CQL implementation:
Equation (24) is incomplete; it is missing the sampling strategy of a_k.
No justification for target networks for offline RL—many offline RL works avoid target networks.
Pseudo-code is incomplete:
Variables ϕ vs φ vs φ– are mixed.
The update frequency d must be clarified.
6. Experimental Setup (Rows 465–474)
Major Concerns
The dataset is insufficiently described.
How many SFC sets?
Number of faulty nodes?
Type of failure patterns?
No baseline for serial migration only.
A comparison between serial and parallel migration is missing, even though serial migration is repeatedly emphasized.
Missing hyperparameters for RL and heuristics.
The topology is fixed to 127 nodes—needs justification.
7. Results and Analysis (Rows 475–547)
Major Issues
Graphs lack confidence intervals.
No statistical significance test (e.g., t-test across 20 runs).
Why does RL-PMO degrade slowly? The explanation is intuitive but not quantified.
The hidden-layer depth experiment is too shallow; missing:
training time,
inference latency,
model size.
Missing sensitivity studies:
Number of candidate nodes k.
Impact of dirty rate.
Impact of bandwidth congestion.
8. Conclusions
Missing future work.
No discussion on:
real-world feasibility,
scalability to thousands of nodes,
limitations of offline dataset collection.
Comments on the Quality of English LanguageNumerous grammar issues (e.g., subject–verb agreement).
Some paragraphs are very long; need splitting.
Figures:
No labels in the text referencing them properly.
Some figures lack axis explanations.
Author Response
Replies to Reviewer 2
Question:
- Title, Abstract, and Keywords
The abstract is overly narrative and does not quantify improvements (e.g., “>15% improvement” must specify the baseline and load level).
“Decision Mamba” appears without explanation; readers unfamiliar with Mamba-based policy networks will not understand its contribution.
The abstract claims that “ensures that learned migration policy can be feasibly executed in real networks,” but feasibility is neither mathematically demonstrated nor experimentally validated.
Response:
We sincerely appreciate the reviewer’s valuable comments. In the revised version of the abstract, we have supplemented the previously missing quantitative information and added a clearer description of the functional role of the Decision Mamba model.
We also thank the reviewer for highlighting this important issue. We acknowledge that the original phrasing in the abstract could have been more precise. The intended meaning of the statement “feasible for execution in real networks” was that the offline reinforcement learning framework is capable of learning effective migration policies from simulation data, and that such policies can subsequently be applied to generate migration decisions in NFV environments. It was not intended to imply that deployment in an actual production network had already been validated. We have revised the manuscript accordingly to clarify this point.
Question:
- Introduction (Rows 35–106)
No problem statement is given in the introduction. The reader does not understand:
What exactly is being optimized?
What defines “success rate”?
What constraints are difficult?
The contributions (Rows 80–106) are high-level and partially overlapping.
Example: Contribution 1 and Contribution 3 both discuss resource-aware decisions.
Missing discussion on why offline RL is preferred over online RL, specifically for SFC migration.
Response:
We sincerely appreciate the reviewer’s important observation. In the revised manuscript, we have added a clearer problem statement, clarified the primary optimization objectives, and highlighted the key challenges on Page 2. In addition, we have refined the logical structure of the “Contributions” section on Page 2 to improve the clarity and completeness of the presentation.
Question:
- Related Work (Rows 107–173)
No comparison table summarizing differences between heuristic, online RL, and offline RL.
The latest 2024–2025 works in offline RL for networking are missing.
The writing is descriptive rather than analytical; it does not explain why earlier methods fail.
Response:
We sincerely appreciate the reviewer’s valuable suggestion. In the revised manuscript, we have strengthened the comparative analysis among heuristic methods, online reinforcement learning, and offline reinforcement learning on Page 3, providing clearer analytical logic and motivation. In addition, on Page 4, we have incorporated recent research advances in applying offline reinforcement learning to networked systems as supplementary context.
Question:
- System Model (Rows 174–310)
This section is the weakest and needs heavy rewriting.
Critical Problems
Notations are inconsistent and sometimes incorrect.
Example: BW notation is used as bw(e) but also B_ij^S.
Indexing of VNF v_j_u_V is unclear.
Constraints (5) and (6) are mathematically incorrect or unreadable.
Summation indices are wrong.
The formula includes unexplained terms (e.g., bw(e_j(u−1)) multiplied by CPU coefficient—why do CPU demands multiply bandwidth?).
Constraint (8) mixes service traffic with migration traffic but does not include routing decisions. The mapping variable µ is defined but not properly integrated.
Dirty rate constraint (9) is conceptually correct but lacks justification (references needed).
Constraint (10) does not account for queueing or processing delays—only link propagation delays.
Evaluation metrics:
Migration time equation (13) is oversimplified and does not properly reflect pre-copy dynamics.
The migration cost formulas (14–16) are not standard and ignore hop count or network distance.
Response:
We sincerely appreciate the reviewer’s detailed and insightful technical observations. In the revised manuscript, we have thoroughly reorganized and refined the system modeling section on Pages 4–8 to ensure that the modeling logic is presented clearly and coherently. Furthermore, we have provided detailed explanations for the physical quantities highlighted by the reviewer and have added the corresponding references to support these definitions.
Question:
- Algorithm Design (Rows 311–464)
No mathematical definition of the state vector length or dimensionality.
The state definition (Equation 20) is unreadable and contains indexing errors.
Missing justification for choosing Decision Mamba instead of Transformer.
DM is a continuous-time state-space model, not a sequential “transformer replacement”. The connection to migration tasks is not clearly explained.
Offline dataset generation lacks details:
How many trajectories?
How many iterations?
What is the reward used for in heuristic algorithms?
What is the distribution of failure types?
The reward function (422) is not defined clearly; the normalization functions n_t, n_s, and n_c are undefined.
The transition probability is not modeled; it is only stated abstractly.
CQL implementation:
Equation (24) is incomplete; it is missing the sampling strategy of a_k.
No justification for target networks for offline RL—many offline RL works avoid target networks.
Pseudo-code is incomplete:
Variables ϕ vs φ vs φ– are mixed.
The update frequency d must be clarified.
Response:
We sincerely thank the reviewer for this valuable comment. In the revised manuscript, we have clarified and strengthened several key components of the methodology:
- State space definition:
The dimensionality of the state space has been clearly defined in Section 4.2. To further improve clarity, we reorganized the state definitions on Page 10 (Section 4.2) and added detailed information about the offline dataset in Section 4.1.
- Justification for using Decision Mamba:
On Page 11 of the revised manuscript, we added a dedicated explanation of the rationale behind selecting the Decision Mamba model and elaborated on its suitability for the SFC migration task.
- Normalization function and sampling strategy for:
We refined the definitions and mathematical formulations of the normalization functions and the sampling strategy forin Sections 4.2 and 4.3 (Pages 11).
- State transition modeling in offline RL:
As clarified in the revised manuscript, in offline reinforcement learning, the state transition probability is implicitly provided by the offline dataset. The model is trained from real transition tuples to learn the value function and the policy , without requiring explicit modeling of the transition kernel.
- Use of target networks:
We employ target networks to enhance training stability. Target networks provide slowly updated targets during Bellman backup operations, which is particularly important when combined with CQL regularization. Furthermore, when using twin critics to mitigate Q-value overestimation, incorporating target networks follows standard practice in modern deep RL algorithms.
- Algorithmic consistency and update frequency :
We have corrected Algorithm 1 (Pages 12) to ensure full notational consistency. Additionally, in Section 4.3 (Page 12), we now explicitly explain the meaning of the update frequency , which controls how often the target networks and critic parameters are updated.
Question:
- Experimental Setup (Rows 465–474)
Major Concerns
The dataset is insufficiently described.
How many SFC sets?
Number of faulty nodes?
Type of failure patterns?
No baseline for serial migration only.
A comparison between serial and parallel migration is missing, even though serial migration is repeatedly emphasized.
Missing hyperparameters for RL and heuristics.
The topology is fixed to 127 nodes—needs justification.
Response:
We thank the reviewer for raising these concerns regarding the experimental setup. In the revised manuscript, we have added the hyperparameters used for both reinforcement learning and heuristic methods in Page 13, Section 5.1. Additionally, detailed information about the offline dataset has been supplemented in Page 10, Section 4.1.
Regarding the comparison between serial and parallel migration, we would like to clarify that the primary contribution of our study focuses on comparing offline RL–based parallel migration with other parallel migration approaches. The theoretical advantages of parallel migration over serial migration—particularly in reducing total migration time—have been well established in prior work. Parallel migration allows multiple VNFs to be transferred simultaneously, whereas serial migration introduces cumulative delays. Therefore, our experiments focus on demonstrating that RL-PMO outperforms existing parallel migration methods in terms of success rate, migration time, and migration cost, rather than revalidating the trade-offs between serial and parallel migration.
Question:
- Results and Analysis (Rows 475–547)
Major Issues
Graphs lack confidence intervals.
No statistical significance test (e.g., t-test across 20 runs).
Why does RL-PMO degrade slowly? The explanation is intuitive but not quantified.
The hidden-layer depth experiment is too shallow; missing:
training time,
inference latency,
model size.
Missing sensitivity studies:
Number of candidate nodes k.
Impact of dirty rate.
Impact of bandwidth congestion.
Response:
We thank the reviewer for the insightful observations. We provide detailed clarifications as follows.
First, all performance curves presented in the figures are plotted based on the mean of five independent experimental runs, each conducted with different random seeds. This ensures the reliability and reproducibility of the reported results. As the figures are mainly intended to illustrate performance trends rather than statistical significance, the comprehensive statistical results of all metrics have been summarized in Table 1 on Page 14 of the revised manuscript. In addition, we have added explicit units to all figures (e.g., migration time in seconds).
We have also included quantitative analyses in Page 13, Section 5.2 of the revised manuscript.
Regarding the comparison of hidden-layer depth, this experiment is designed to evaluate the trade-off between model capacity and deploy ability in edge-network environments. The goal is to assess how different depths of the Decision Mamba policy network affect migration performance. The detailed model configuration and architecture specifications of Decision Mamba have been provided on Page 13, Section 5.1.
In response to the reviewer’s concern on the sensitivity to the number of candidate nodes , we have added experiments with explicit variation of , and the corresponding results are summarized in Page 15, Section 5.2.
Regarding the sensitivity analysis for dirty-page rate and bandwidth congestion, we clarify that in our setting, different VNF types have different resource coefficients and traffic scaling factors. This causes substantial variation in migration data volume and effective dirty-page behavior across VNFs and across scenarios. Under such randomized and diverse deployment patterns, even with fixed link bandwidth and fixed dirty-page rate, the actual load levels, congestion severity, and migration difficulty vary significantly across experimental runs. Therefore, introducing additional variations in dirty-page rate or link bandwidth would not meaningfully change the system behavior or lead to new insights, as the primary source of variability is already dominated by VNF heterogeneity and stochastic deployment patterns.
Question:
- Conclusions
Missing future work.
No discussion on:
real-world feasibility,
scalability to thousands of nodes,
limitations of offline dataset collection.
Response:
We sincerely appreciate the reviewer’s valuable suggestion. In the revised manuscript, we have expanded Section 6 by adding a more thorough discussion on practical feasibility and scalability. We also explicitly outlined several directions for future work to provide a clearer perspective on the potential extensions of our study.
Round 2
Reviewer 1 Report
Comments and Suggestions for Authors
Thank you for considering my comments.
Comments on the Quality of English LanguageIt is not needed to perform a thorough language edit.
Author Response
We sincerely appreciate your careful review of our manuscript and your valuable suggestions. Thank you for pointing out the need to more clearly articulate the relevance of our work to sensor technologies and sensor networks. We have revised the manuscript accordingly, and a summary of the major modifications is provided below.
- Revisions to Section 1 (Introduction)(Pages 1–2)
To address your request for a clearer connection between our research and sensor-network applications, we added new explanations in the Introduction that explicitly clarify the following points:
- In modern IoT and sensor-network scenarios, large volumes of real-time sensing data must be processed by Virtual Network Functions (VNFs) deployed at edge nodes, highlighting the critical role of Service Function Chains (SFCs) in such systems.
These revisions clarify that the failure-driven SFC migration problem studied in this work is inherently aligned with the operational characteristics and reliability requirements of sensor networks. By integrating this contextual explanation directly into the Introduction, we ensure that the motivation and significance of our study more clearly fit within the scope of the journal.
- Additions to Section 2(Related Work)(Pages 3–4)
Following your suggestions, we incorporated additional sensor-network considerations into Section 2. Specifically:
- We emphasize that sensor networks exhibit highly dynamic traffic patterns and resource usage, making it more difficult for heuristic algorithms to model VNF interactions and avoid local optima.
- We further highlight that online reinforcement learning is unsuitable for failure-driven SFC migration in sensor-network environments with stringent real-time requirements, as unsafe exploratory actions and slow convergence cannot be tolerated in such settings.
These additions strengthen the justification for our proposed offline reinforcement learning approach, demonstrating that it is particularly well suited to the strict timing, reliability, and stability requirements of SFC migration in sensor networks.
This directly responds to the editorial request by showing the applicability and necessity of our method in sensor-based systems.
- Revisions to Section 6 (Conclusions)(Page 18)
To further emphasize the relevance of our contributions, we added the following statement to the Conclusion:
- The robustness, stability, and high migration success rate of RL-PMO make it especially suitable for sensing and IoT-edge environments, where stable and successful SFC migration is essential for maintaining the reliability of critical sensing applications.
This addition clearly articulates the broader significance of our method for sensor-network scenarios, aligning the manuscript more closely with the journal’s scope.
Through the revisions to Sections 1, 2, and 6, we have ensured that the manuscript now presents a clear and coherent connection between our research and sensor technologies as well as sensor-network applications, fully addressing the concerns raised by the editors. We are sincerely grateful for your guidance, which has helped us enhance the clarity and relevance of our work.
Reviewer 2 Report
Comments and Suggestions for Authors
The authors addressed all of my concerns, no more comments.
Author Response

(The authors gave the same response as above.)
